# Impact of the ABxSG Mobile Application on Antibiotic Prescribing: An Interrupted Time Series Study

**DOI:** 10.3390/antibiotics14090933

**Published:** 2025-09-16

**Authors:** Lai Wei Lee, Shena Yun Chun Lim, Yvonne Peijun Zhou, Shimin Jasmine Chung, De Zhi Chin, Andrea Lay Hoon Kwa, Winnie Hui Ling Lee

**Affiliations:** 1Division of Pharmacy, Singapore General Hospital, Outram Road, Singapore 169608, Singapore; shena.lim.y.c@sgh.com.sg (S.Y.C.L.); yvonne.zhou.p.j@sgh.com.sg (Y.P.Z.); andrea.kwa.l.h@sgh.com.sg (A.L.H.K.); 2Department of Infectious Diseases, Singapore General Hospital, Outram Road, Singapore 169608, Singapore; jasmine.chung.s.m@singhealth.com.sg; 3Office of Value Based Healthcare, Singapore General Hospital, Outram Road, Singapore 169608, Singapore; chin.de.zhi@sgh.com.sg; 4SingHealth Duke-NUS Medicine Academic Clinical Programme, 8 College Road, Singapore 169857, Singapore; 5Emerging Infectious Disease Program, Duke-NUS Medical School, 8 College Road, Singapore 169857, Singapore

**Keywords:** antimicrobial stewardship, antibiotic use, mobile application, behavioral design thinking

## Abstract

Background: A point prevalence survey conducted at Singapore General Hospital in 2021 showed 48% of inpatients on antibiotics. We hypothesize that a mobile application, transforming complex antibiotic prescribing information into a succinct and individualized resource, will empower healthcare professionals and improve antibiotic prescriptions. Hence, we developed ABxSG using the behavioral design thinking approach (BDTA) to ensure positive user experience and sustained engagement. We aim to evaluate the impact of ABxSG on the proportion of inpatients on antibiotics, antibiotic appropriateness, and the number of antibiotic-related interventions by pharmacists. Methods: ABxSG was launched in March 2023. An interrupted time series analysis was conducted to evaluate its impact on the above outcomes measured using data collected from October 2021 to September 2024. There were 18 data points pre- and post-ABxSG. Results: Following the ABxSG launch, there was an immediate reduction in the proportion of inpatients on antibiotics by 1.66% (*p* < 0.01), followed by a sustained reduction of 3.12% at 18 months (*p* < 0.01). Piperacillin-tazobactam appropriateness increased by 2.76% at 1 month (*p* = 0.11) and further increased by 7.09% at 18 months (*p* < 0.05). Similarly, ceftriaxone appropriateness increased by 5.74% (*p* < 0.05) at 1 month and remained above pre-ABxSG levels. There was a significant increase in the number of pharmacist-led interventions for dosing optimization, with 96 more interventions/month at 18 months (*p* = 0.14). Conclusion: Antimicrobial stewardship teams must remain agile, embrace innovations, and adopt digital technologies to engage and empower clinicians. ABxSG reduced the proportion of inpatients on antibiotics and improved antibiotic prescriptions. The incorporation of BDTA in ABxSG, strong hospital leader support, and strategic planning to promote adoption led to its success.

## 1. Introduction

Antimicrobial stewardship programs (ASPs) are advocated in all healthcare institutions [1,2], as they have been shown to reduce antibiotic consumption and healthcare costs and slow development of antimicrobial resistance without compromising patient safety [3,4,5,6]. In 2008, Singapore General Hospital (SGH) established an Antimicrobial Stewardship Unit (ASU), comprising infectious disease physicians and clinical pharmacists, employing multi-pronged approaches, e.g., prospective audit and feedback (PAF), antibiotic guideline development, and ongoing physician education and engagement, primarily targeting inappropriate prescribing of broad-spectrum intravenous (IV) antibiotics (carbapenems and piperacillin-tazobactam). Despite achieving good outcomes in terms of reduced length of hospitalization and healthcare cost for ASU-audited patients [3,4], a point prevalence survey (PPS) conducted using Global-PPS methodology in 2021 showed that 48% of inpatients received at least one antibiotic on the day of the survey [7]. This prevalence is significantly higher than in other developed countries [such as Europe (25.9–39.5%), North America (36.9%), and East and South Asia (38.5%)] [8]. Furthermore, about 20–25% of antibiotic prescriptions at SGH are inappropriate based on in-house observations.

Notably, the PPS in 2021 also showed that 55% and 14% of all inpatient antibiotic prescriptions at SGH are initiated for empiric treatment and surgical prophylaxis, respectively. Antibiotic prescribing guidelines are, therefore, pertinent in driving appropriate antibiotic prescribing for empiric treatment and surgical prophylaxis [9,10,11]. At SGH, even though these guidelines are hosted on the institution intranet, there are only ~5% (63/1400 doctors) unique visitors to the webpage daily (in-house data). Poor traffic could be partly due to access limitations via hospital systems as well as users’ difficulty in navigating the intranet. A survey conducted by Pelullo et al. showed that surgeons who used unnecessary surgical prophylaxis were concerned about patients developing surgical site infections if antibiotics were not given [12]. Hence, apart from ensuring guideline accessibility, the availability of effective learning resources and motivating clinicians to refer to them are important factors in advocating appropriate antibiotic prescribing. In Singapore, where smartphones are ubiquitous, we proposed to develop a mobile application as an antimicrobial stewardship tool that empowers physicians and pharmacists to use antibiotics optimally. This idea was supported by a survey we conducted with 50 physicians of all seniority levels from both medical and surgical disciplines. All respondents preferred to access guidelines and learning resources on their smartphones; 97.5% expressed that they would prescribe antibiotics more confidently with a mobile application.

With the use of mobile applications gaining increasing popularity worldwide, a systematic review by Alkhaldi et al. evaluating the use of mobile health applications showed that only half of the studies reported changes in medication prescribing habits and a quarter showed improvement in knowledge among prescribers [13]. Hence, to maximize users’ adoption and ensure meaningful impact on antimicrobial prescribing, we adopted the behavioral design thinking approach (BDTA) in the development of the mobile application. Voorheis et al. described BDTA as a five-step amalgamation of best practices from behavioral design and design thinking [14]. The five steps are (1) empathize with users and their behavior change needs; (2) define user and behavior change requirements; (3) ideate user-centered features and behavior change content; (4) prototype a user-centered solution that supports behavior change; and (5) test the solution against users’ needs and for its behavior change potential (Figure 1).

On 27 March 2023, we launched ABxSG, Singapore’s first antimicrobial mobile application. ABxSG is a customized mobile application, designed and built in-house for SingHealth institutions, to empower healthcare professionals (HCPs) by consolidating and transforming complex antibiotic prescribing information into a succinct and individualized resource with the ultimate goal of promoting antimicrobial stewardship. ABxSG can be downloaded for free by all HCPs from all participating SingHealth hospitals via both the Apple App Store and the Google Play Store. A series of roadshows were conducted with major medical and surgical departments from April to June 2023 to promote download and use of ABxSG. Additional promotional materials, which included an intranet banner (Appendix A), a computer screensaver (Appendix A), and a short video clip demonstrating the key features of ABxSG, were disseminated and integrated into various HCP orientation programs to increase awareness. A physical publication booth was also set up within the hospital to promote downloads and address any queries.

Here, we aim to evaluate the impact of ABxSG on the (1) proportion of inpatients on antibiotics; (2) appropriateness of antibiotic use; and (3) number of interventions made by pharmacists to improve antibiotic prescribing.

## 2. Results

As of September 2024, there were 1902 downloads of ABxSG; 70% of the users were doctors, followed by pharmacists (28%) and nurses (2%). At SGH, antibiotics can only be prescribed by doctors, which makes them the primary target audience for ABxSG. However, as pharmacists and nurses are also involved in the care of patients on antibiotics, the app also includes information that aids those healthcare professionals.

### 2.1. Proportion of Inpatients on Antibiotics

The proportion of inpatients on antibiotics was rising at 0.17%/month pre-launch and slowed down post-launch of ABxSG (*p* < 0.05) (Table 1, Figure 2). Upon ABxSG launch, there was a significant immediate level drop of 1.66% at 1 month (*p* < 0.01) and 2.09% at 6 months (*p* < 0.01) in the proportion of inpatients on antibiotics, compared to the predicted trend without ABxSG (based on the pre-slope trend). The reduction was sustained with a decrease in inpatients on antibiotics by 2.6% at 12 months (*p* < 0.01) and 3.12% at 18 months (*p* < 0.01) compared to the predicted pre-slope trend.

### 2.2. Proportion of Appropriate Use of Audited Antibiotics

There was an immediate increase in appropriate use of piperacillin-tazobactam by 2.76% at 1 month (*p* = 0.11) and continued to significantly increase to 4.04%, 5.56%, and 7.09% at 6, 12, and 18 months post-launch (*p* < 0.05) (Table 1, Figure 3A). Similarly, for ceftriaxone prescriptions among surgical departments, the proportion of appropriate ceftriaxone use increased immediately by 5.74% (*p* < 0.05) at 1 month post-launch, and appropriateness was sustained till 18 months (Figure 3B).

The proportion of appropriate meropenem use did not change in trend pre- and post-launch of ABxSG (*p* = 0.50) (Figure 3C).

### 2.3. Number of Pharmacist-Led Interventions

At SGH, all inpatient antibiotic orders are verified by pharmacists before they are administered to the patients, and the continued use of antibiotics is regularly reviewed by either ward pharmacists or ASP pharmacists. If the antibiotics are deemed inappropriately used at any time point, an intervention will be made by a pharmacist to modify the antibiotic order. At 1 month post-launch, the number of interventions for antibiotic choice optimization was immediately reduced by 20 interventions per month (*p* = 0.08) (Table 1, Figure 4A). Compared with predicted number of interventions based on the pre-launch trend, there was a reduction of 27 interventions at 6 months (*p* < 0.05), 36 interventions at 12 months (*p* < 0.05), and 45 interventions at 18 months (*p* < 0.05) post-launch.

In contrast, there was a steeper increase in the number of pharmacist-led interventions for dosing regimen optimization post-launch of ABxSG (Figure 4B). At baseline, there was an increase of 9 interventions for dose regimen recommendations every month. At 1 and 6 months after the ABxSG launch, the number of such interventions was approximately 10 interventions (*p* = 0.77) and 35 interventions (*p* = 0.35) more per month, respectively, compared to the predicted numbers based on the pre-launch trend. The rise in slope post-launch resulted in an increase of 65 interventions (*p* = 0.18) and 96 interventions (*p* = 0.14) per month at 12 and 18 months, respectively, post-launch (Figure 4B).

## 3. Discussion

### 3.1. ABxSG Reduces the Proportion of Patients on Antibiotics

Apart from an observable drop in the proportion of patients on antibiotics soon after the introduction of ABxSG, we also noted that the monthly rate of increase had slowed significantly up to 18 months post-launch (*p* < 0.05), leading to a significant reduction of 3.1% (*p* < 0.05) (Figure 2). This 3.1% reduction translates to approximately 650 more antibiotic-free days per month, based on a mean of 21,000 patient-days-on-antibiotics per month pre-launch of ABxSG. Reduction in antibiotic utilization is expected to retard emergence of resistant bacteria, as well as reduce length of hospitalization and facilitate better healthcare resource (e.g., hospital beds and healthcare manpower) allocation [15,16]. Additionally, direct savings on healthcare and manpower costs are expected. Based on in-house data where an estimated 60% of inpatients on antibiotics are prescribed IV formulation, we project that the monthly IV drug administration cost [approximately SGD 78/day (~USD 60)] avoided with 650 antibiotic-free days is more than SGD $30,000 (~USD 23,000). Furthermore, assuming 30 min is needed to prepare and administer an IV antibiotic, an estimated 585 h/month of nursing time can be saved [approximately SGD $17,000 (~USD 13,000) a month].

### 3.2. ABxSG Improved the Appropriateness of Piperacillin-Tazobactam and Ceftriaxone

At SGH, piperacillin-tazobactam and ceftriaxone are two of the most prescribed IV antibiotics. At 18 months post-launch, there was an improvement in the appropriateness of piperacillin-tazobactam and ceftriaxone by 7.1% and 2.1%, respectively, compared to the predicted value based on the pre-launch trend (Figure 3A,B). While this improvement could have been driven in part by the heightened awareness of antibiotic stewardship from the various publicity activities conducted to launch ABxSG, we believe that the provision of readily available institution guidelines and learning resources reinforced evidence-based practice and guideline adherence, which contributed to the sustained effect on appropriateness even at 18 months post-launch. For example, doctors are now made aware that ceftriaxone use should not be extended post-surgery as surgical prophylaxis or that they should prescribe piperacillin-tazobactam only for nosocomial infections. This observation was also reported in other centers with similar interventions [17,18]. While there was no significant change in trend with meropenem, overall appropriateness of meropenem remained high above 80% throughout the study period. At SGH, more than 90% of meropenem prescriptions are appropriately initiated, typically prescribed in hemodynamically unstable patients with severe infections. However, the challenge of appropriate meropenem use resides in the reluctance to de-escalate promptly, and our study highlights that behavioral change is indeed an uphill battle [13,19]. Perhaps, behavioral change regarding antibiotic use in “sicker” patients involves strategies beyond the introduction of an antimicrobial mobile application.

Although appropriateness data is only available for selected broad-spectrum antibiotics (such as piperacillin-tazobactam, ceftriaxone, and meropenem) audited by ASU, the reduction in the number of pharmacist interventions on antibiotic choice for all antibiotic types post-ABxSG launch suggests that appropriate antibiotic use extended beyond these studied antibiotics. At 18 months, the number of interventions made was 45 fewer than predicted (Figure 4A). A systematic review by Helou R et al. corroborated our study findings that antibiotic mobile applications improved knowledge of prescribers and reduced prescribing errors, especially among junior ordering physicians [20].

### 3.3. ABxSG Empowered Pharmacists to Optimize Antibiotic Doses

At SGH, prescribers are generally more focused on selecting the antimicrobial and heavily depend on pharmacists to optimize antimicrobial dosing regimens. Interestingly, although not statistically significant, the increase in the number of pharmacist interventions pertaining to drug dosing optimization post-ABxSG launch is apparent (Figure 4B). With better access to drug monographs highlighting dosing regimens for specific indications or patient populations (including patients who are on dialysis settings specific to the institution’s practice), we postulate that pharmacists are now more empowered to identify suboptimal dosing regimens without needing to check several other references and can intervene more confidently to optimize therapy.

### 3.4. Success of ABxSG

In our study, we have demonstrated that ABxSG is an effective antimicrobial stewardship tool, which improved antibiotic prescribing. We attribute the success of ABxSG to (1) integration of an effective BDTA in the design and development of the mobile application; (2) unwavering support from hospital senior leaders; and (3) effective promotion of the mobile application on various platforms.

Using BDTA, we ensured that in-depth research was conducted to empathize with the needs of our local users by employing user-centric surveys. We also conducted a literature search that evaluated existing mobile applications that promoted antimicrobial stewardship overseas and identified areas for improvement. For example, Roganovic et al. reported that their application did not consider follow-up measures after initial antibiotic recommendation, and content was angled towards only dentists and not other HCPs [21]. de Lorenzi-Tognon et al. suggested that their application recommendations were not individualized to local epidemiology and antimicrobial resistance patterns [22]. Hence, we ensured that the antibiotic guidelines contained not only recommendations on empiric antibiotic initiation but also de-escalation options and clinical pearls on managing various infective conditions. Furthermore, we also integrated features that were beneficial to both pharmacists and nurses. For example, we ensured that administration information and monitoring parameters are made available for nurses, whereas dosing specific to indication and patient population is made available for pharmacists. Additionally, we ensured that we integrated up-to-date information that is individualized based on institution practice, local epidemiology, and antimicrobial resistance patterns. Importantly, we incorporated a well-structured and intuitive flow into the user interface, which is aligned to what a prescriber would consider when prescribing antibiotics. For example, the prescriber would use medical calculators to derive infection severity, then refer to corresponding guidelines for empiric antibiotic recommendations, followed by referring to individual drug monographs for specific monitoring parameters, drug–drug interactions, and dosing regimens.

Apart from focusing on content pertaining to the management of infective conditions and antimicrobial use, immense effort was made during the brainstorming stage by ASU to incorporate features that optimize user experience and encourage sustained user engagement. One key feature is the introduction of a simple-to-use user interface. Information is thematic and separated into intuitive sections, ensuring that even first-time users can easily navigate through the application. The inclusion of a search function enhanced the accessibility of the mobile application, allowing users to search for information using keywords. The bookmarking feature allowed quick access to frequently accessed pages, removing the need to plow through the mobile application to look for the same pages when they relaunch ABxSG. With the advent of social media and messaging applications, a share function was incorporated, allowing users to share pages with other users through the various messaging applications as long as they had ABxSG installed. ABxSG was also developed with both of the primary mobile operating systems in mind, ensuring that almost all HCPs would be able to benefit from the application, even though cross-platform mobile application development would increase the time taken and difficulty of mobile application development.

Another key feature for a successful mobile application is the provision of regular updates. The integrity of the information presented in the mobile application is kept updated, with frequent revisions should new primary evidence or workflow be introduced. For example, information on emerging infectious diseases such as mpox and COVID-19 was consistently being updated when new information was available. During pandemic situations, this mobile application can serve as another platform where rapidly changing guidelines or workflows can be disseminated. We also included a “feedback” section that allows users to offer constructive suggestions to further improve the application. Lastly, we also enticed institution HCPs to download ABxSG by the inclusion of a call list, where users can contact various departments (for example, the pharmacy and microbiology lab) using one click conveniently.

Apart from developing a mobile application that provides comprehensive, individualized, and user-friendly content, garnering support from hospital leaders was critical in supporting our new antimicrobial stewardship strategy [2]. Unwavering support and buy-in from senior management enabled both ongoing funding disbursement and manpower deployment required for the development and maintenance of ABxSG. Additionally, their support provided the green light for ASU to freely promote ABxSG on various platforms, including hospital intranet, screensavers, ward rounds, physical publication booths, and roadshows to major clinical departments. Importantly, we encouraged physicians to recommend ABxSG to junior doctors to improve uptake, which was also shown to be an important factor for young HCPs to adopt the use of new mobile health applications [23]. Notably, ABxSG was well received with a total of 680 users at 3 months, which later grew to 1902 users at 18 months.

### 3.5. Study Limitations

Our study has several limitations. Firstly, the use of ITS with ARIMA using 18 time points pre- and post-ABxSG launch may not be able to fully account for variability in workforce deployment cycles (e.g., new junior doctors who are rotated into the hospital at different time points of the year may be unaware of ABxSG or not familiar with institution guidelines for appropriate antibiotic prescribing). However, we reduced the impact of this problem by ensuring the inclusion of 2 critical data points (July 2023 and July 2024) post-ABxSG, which coincides with the entry of new junior doctors into the hospital. We acknowledge that a longer time period would be necessary to determine whether the positive impact of ABxSG on antibiotic prescribing was truly sustainable.

Secondly, the concurrent implementation of other concurrent antimicrobial stewardship initiatives could have also contributed to the sustained effects of a lowered proportion of patients on antibiotics and increased appropriateness of antibiotic use. For example, we had stepped up engagement with surgical departments during the study period, with initiatives such as revision of guidelines and collaborating in quality improvement projects to curb unnecessary use of antibiotics, especially in the area of surgical prophylaxis. Nonetheless, our results showed an immediate significant reduction in the proportion of inpatients on antibiotics following the ABxSG launch, which we could confidently attribute to the impact of ABxSG.

Thirdly, the appropriateness of antibiotics was not assessed based on indication, as data pertaining to smaller groups are susceptible to fluctuations due to small denominators and hence unlikely to identify any meaningful trends. Nonetheless, we noticed that there were no major differences in the type of antibiotic inappropriateness (based on choice, duration, route, dosing, and the need for antibiotics) over time, and we believe that improvement in antibiotic prescribing likely applies to all categories of indications in general.

Lastly, due to limitations in the application analytics, we are unable to accurately identify the department that users belong to and their frequently visited pages. We acknowledge that understanding the user profile and usage patterns is critical in improving the mobile application. Hence, one important lesson learned for us would be the importance of building a robust application analytical system following any mobile application development to facilitate ongoing enhancement.

## 4. Methods

### 4.1. Developing ABxSG Using the Behavioral Design Thinking Approach

In step 1 of BDTA, we conducted a user-centric survey with institution HCPs to understand challenges in antimicrobial prescribing, prescribing behaviors/culture, and features that were deemed beneficial or lacking in existing commercially available mobile applications. Additionally, we also sought to understand the limitations and challenges in implementing mobile applications that aimed to promote antimicrobial stewardship overseas from the primary literature. We then analyzed the needs of HCPs and defined the major themes required in the application of behavioral change in step 2. Following which, brainstorming of specific application features along with user interface/user experience design was performed with the ASU pharmacists and physicians in step 3. Thereafter, a prototype was developed and tested with a small group of HCPs, undergoing iterative refinements to enhance user experience prior to the official launch of the application. The major features of ABxSG include (1) institution-specific prescribing guidelines; (2) drug monographs; (3) medical calculators; and (4) a learning corner (Figure 5). To allow for content from multiple institutions of the SingHealth hospitals to be hosted distinctly within the same application, we built a containerized infrastructure so that our sister institutions can upload their personalized content in their respective “containers”. This feature supports physicians with cross-institution practices, allowing them to switch “containers” (institutions) easily within the same application.

Unwavering support and buy-in from senior management enabled both funding disbursement [SGD $60,000 (~USD 46,000) for developing ABxSG] and manpower deployment required for the development and maintenance of ABxSG.

#### 4.1.1. Institution-Specific Prescribing Guidelines

There is a total of 98 institution-specific guidelines found in SGH ABxSG (42 treatment and 56 prophylaxis guidelines). Each guideline incorporates evidence-based recommendations, customized to local antibiograms and practices. Antibiotic recommendations are also available for patients with penicillin allergy and positive methicillin-resistant *Staphylococcus aureus* colonization status. The application also suggests the duration of therapy and the type of step-down therapy to consider should patients get better and whose workup did not garner any positive microbial cultures.

#### 4.1.2. Drug Monographs

ABxSG differentiates itself from commercially available applications by introducing drug monographs of institution formulary antimicrobials that are designed to answer institution-specific concerns. For example, it contains information on antimicrobial dosing tailored to in-house treatment protocols and dialysis modalities. Special instructions on brand-specific recommendations on dilution and administration instructions, especially for fluid-restricted patients, are also included. Additionally, the drug monographs include institution-specific practices, such as formulary restriction status and suitability for outpatient parenteral antimicrobial therapy. Furthermore, a special blue box in the monograph highlights crucial information pertaining to therapeutic drug monitoring (TDM) or monitoring of adverse drug reactions and drug–drug interactions.

#### 4.1.3. Medical Calculators

ABxSG contains a curated list of medical calculators aimed at improving antimicrobial prescribing and are commonly used by SGH doctors and pharmacists. For example, body habitus and renal function calculators aim to guide dosing in obese and renally impaired patients, respectively, while disease-specific calculators (such as the CURB-65 score for bacterial pneumonia and the ISARIC-4C score for COVID-19) determine infection severity and thus, guide appropriate antibiotic selection from the prescribing guidelines.

Of note, ABxSG includes an in-house vancomycin TDM calculator that has been validated in our patient population. Beyond initial vancomycin dose (both loading and maintenance) recommendations, the vancomycin TDM calculator also suggests dosing adjustments based on measured trough levels. Designed to support clinical decision-making, this calculator enables doctors and pharmacists to titrate vancomycin doses with greater confidence and autonomy.

#### 4.1.4. Learning Corner

The learning corner provides a one-stop location for clinical pearls for the management of common infectious disease conditions. It provides key, bite-sized recommendations from international guidelines and landmark clinical trials, enabling prescribers to remain up to date with the latest evidence, strengthening their ability to (1) more accurately diagnose, with consequent decisions on antibiotic initiation; (2) institute relevant infection control practices; and (3) rationalize antibiotic choice, duration, or dosing.

### 4.2. Study Design

In this single-center time-series study, all patients who were hospitalized in SGH (a 2000-bed quaternary-care hospital) from October 2021 to September 2024 were included. The 3-year period was chosen to allow comparison of an 18-month pre-launch (October 2021 to March 2023) versus an 18-month post-launch (April 2023 to September 2024) of ABxSG.

### 4.3. Information Collected

Data pertaining to the monthly proportion of inpatients on antibiotics was obtained from the institution data repository warehouse and was derived from dividing the total number of days where patients were prescribed at least one antibiotic by the total number of inpatient days. The proportion of inpatients on antibiotics, as a metric, encompasses all antibiotic types and is independent of dosing variations among different patient populations. When trended serially over time, it would be a more suitable metric (than defined daily doses or days of therapy) when evaluating the impact of an antimicrobial stewardship strategy on all antibiotics. Additionally, this measure is one of the key performance indicators reportable to SGH’s senior management and Singapore’s Ministry of Health. Antibiotic appropriateness was obtained from the ASU’s PAF database. In PAF, ASU pharmacists review prescriptions for selected broad-spectrum antibiotics (e.g., meropenem and piperacillin-tazobactam use hospital-wide and ceftriaxone use in surgical departments) and provide prescribers with recommendations when antibiotics are deemed inappropriate. Antibiotic overall appropriateness is determined at initiation and upon antibiotic discontinuation based on appropriate choice, dose, route, duration, and/or if use of an antibiotic was even indicated.

### 4.4. Outcomes

The primary objective of the study is to evaluate the impact of ABxSG on the proportion of inpatients on antibiotics over time. The secondary objectives include evaluating the impact of ABxSG on the appropriateness of antibiotics (meropenem, piperacillin-tazobactam, and ceftriaxone) and the number of interventions made by pharmacists to improve antibiotic prescribing over time. Notably, meropenem, piperacillin-tazobactam, and ceftriaxone contribute to 90% of audits conducted by ASU pharmacists and are important workhorses in the antibiotic armamentarium, whose susceptibility must be preserved. Based on aggregated data from acute care hospitals in Singapore in 2021, resistance to ceftriaxone ranges from 21.0% for *Escherichia coli* to 23.3% for *Klebsiella pneumoniae*. For meropenem, resistance ranges from 0.3% for *Escherichia coli* to 31.5% *for Acinetobacter baumanii* [24].

### 4.5. Statistical Analysis

An interrupted time series (ITS) analysis, based on an autoregressive integrated moving average (ARIMA) model fitted to the observed data of the aforementioned primary and secondary outcomes and adjusted for first-order autocorrelation, was conducted to evaluate the impact of ABxSG over time [25,26]. The analysis included 18 data points before and after the launch of ABxSG to determine its effect on the test variable (≥8 data points were required for an accurate ITS) [27]. Pre-slope was the trend observed prior to ABxSG, whereas slope change refers to the change in direction and gradient of the trend line after the ABxSG launch. The level effect at 1, 6, 12, and 18 months post-launch was the difference between the observed and predicted value based on trending from the pre-launch period.

All statistical analyses were performed using IBM SPSS Statistics for Windows, Version 20.0 (IBM Corp., Armonk, New York, NY, USA).

## 5. Conclusions

Antimicrobial stewardship teams must be agile, embrace innovations, and readily adopt digital technologies to sustainably engage, educate, and empower clinicians.

The ABxSG mobile application provided ready access to knowledge that empowered HCPs (both physicians and pharmacists) to use antibiotics optimally. Its use in our hospital was associated with improved appropriateness of antibiotic use with a reduction in the proportion of inpatients on antibiotics. Incorporation of BDTA principles in the design and development of ABxSG was critical in ensuring positive user experience and sustained user engagement. Additionally, strong support from hospital leaders along with strategic planning on promoting its use within the institution led to its successful adoption and good outcomes. ABxSG will continue to expand its content to enhance its capabilities in providing customized information that is adapted to local practice to drive antimicrobial stewardship efforts to combat rising bacterial resistance and improve patient outcomes.

## Figures and Tables

**Figure 1 antibiotics-14-00933-f001:**
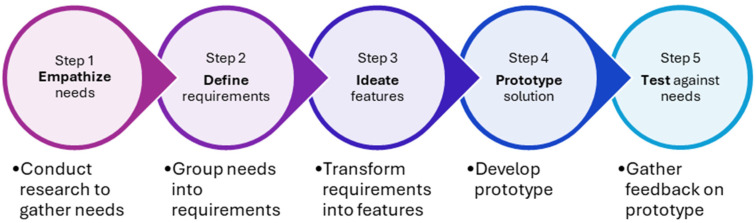
Flowchart of the behavioral design thinking approach.

**Figure 2 antibiotics-14-00933-f002:**
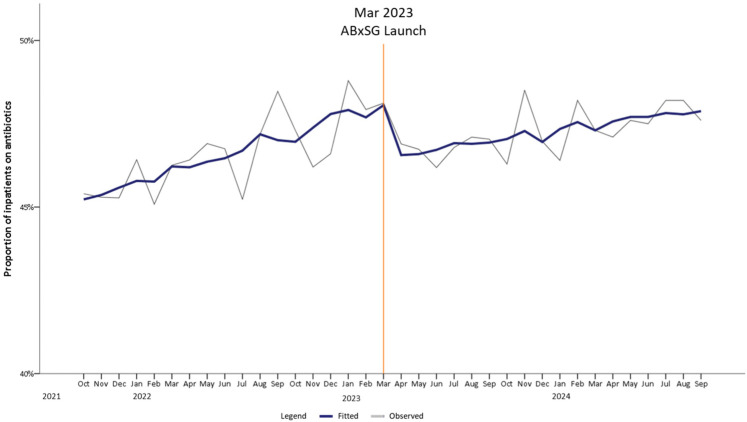
Trend of proportion of inpatients on antibiotics pre- and post-launch of ABxSG (October 2021 to September 2024).

**Figure 3 antibiotics-14-00933-f003:**
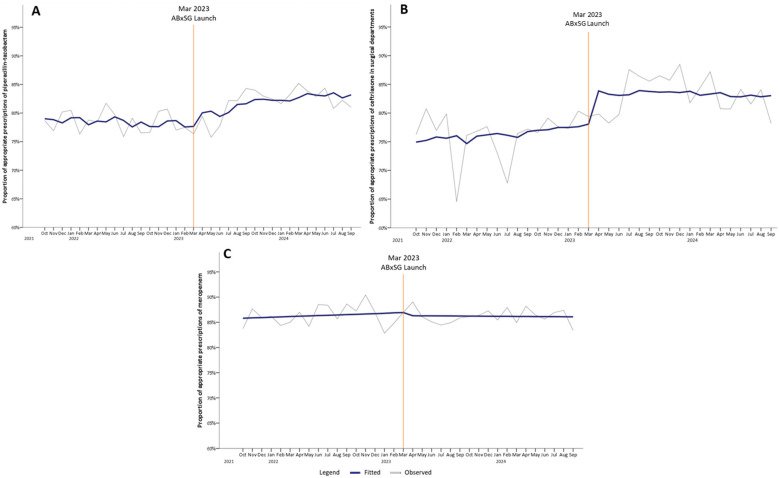
Proportion of appropriate prescriptions of audited antibiotics, namely piperacillin-tazobactam (**A**), ceftriaxone (**B**), and meropenem (**C**), pre- and post-launch of ABxSG (October 2021 to September 2024).

**Figure 4 antibiotics-14-00933-f004:**
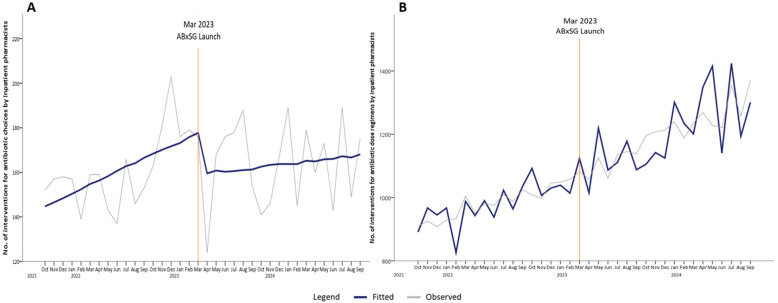
Number of interventions for antibiotic choices (**A**) and dose regimens (**B**) by inpatient pharmacists’ pre- and post-launch of ABxSG (October 2021 to September 2024).

**Figure 5 antibiotics-14-00933-f005:**
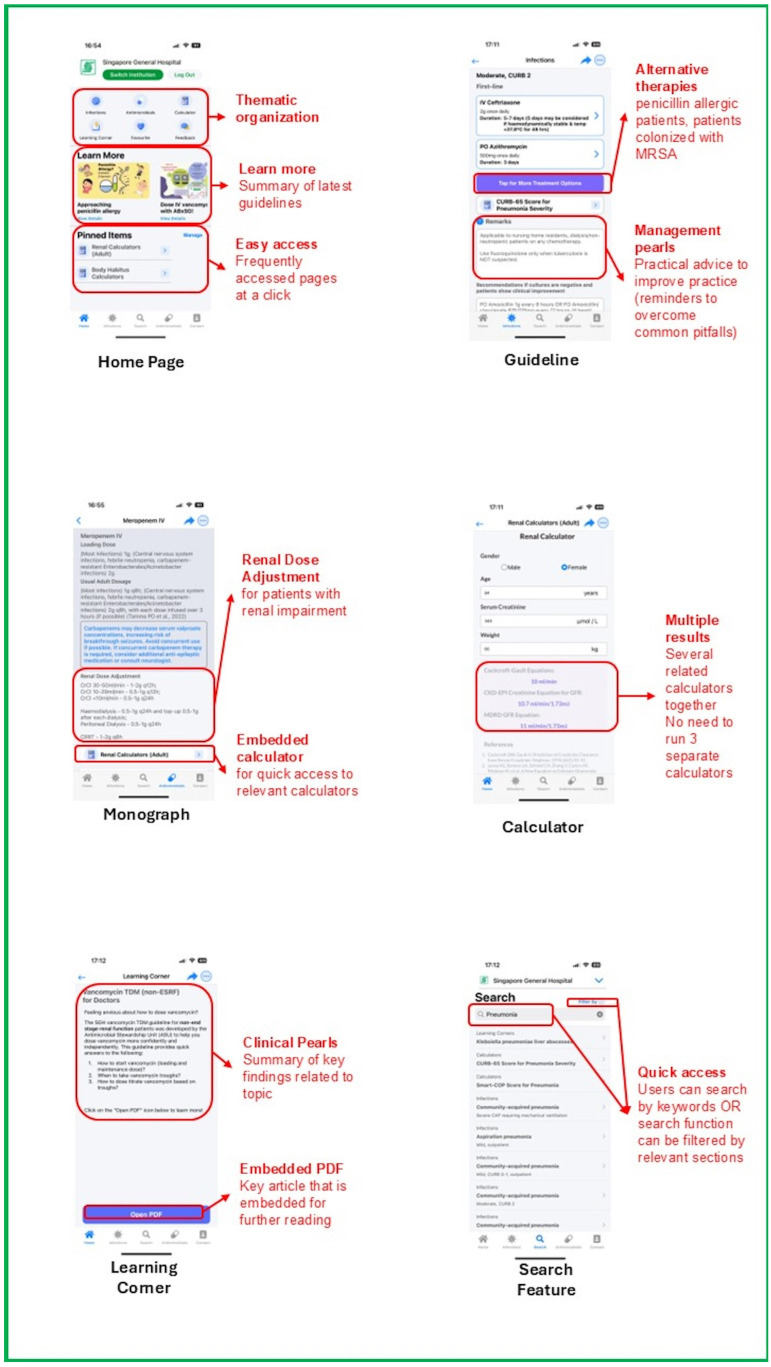
Snapshots of ABxSG.

**Table 1 antibiotics-14-00933-t001:** Results from Interrupted Time Series ^†^ Change in slope refers to the change in direction and gradient of the trend line after the ABxSG launch. ^‡^ Appropriateness is evaluated by the antimicrobial stewardship unit via prospective audit feedback on hospital-wide piperacillin-tazobactam and meropenem, as well as ceftriaxone in surgical departments only.

	Pre-Slope	Change in Slope ^†^	Level Effect at Respective Timepoints			
			1 Month	6 Months	12 Months	18 Months
**Proportion of inpatients on antibiotics (%)**	0.17 (*p* < 0.01)	−0.09 (*p* < 0.05)	−1.66 (*p* < 0.01)	−2.09 (*p* < 0.01)	−2.60 (*p* < 0.01)	−3.12 (*p* < 0.01)
**Proportion of appropriate prescriptions of antibiotics ^‡^**						
Piperacillin-tazobactam (%)	−0.07 (*p* = 0.53)	0.26 (*p* = 0.11)	2.76 (*p* = 0.11)	4.04 (*p* < 0.05)	5.56 (*p* < 0.05)	7.10 (*p* < 0.05)
Ceftriaxone (%)	0.17 (*p* = 0.39)	−0.21 (*p* = 0.44)	5.74 (*p* < 0.05)	4.66 (*p* = 0.15)	3.38 (*p* = 0.43)	2.09 (*p* = 0.71)
Meropenem (%)	0.065 (*p* = 0.43)	−0.08 (*p* = 0.50)	−0.68 (*p* = 0.57)	1.01 (*p* = 0.43)	−1.54 (*p* = 0.39)	−2.01 (*p* = 0.40)
**Number of antibiotic-related interventions by pharmacists**						
Drug choice	1.94 (*p* < 0.05)	−1.46 (*p* = 0.18)	−20.24 (*p* = 0.08)	−27.56 (*p* < 0.05)	−36.34 (*p* < 0.05)	−45.11 (*p* < 0.05)
Dose regimen	9.12 (*p* < 0.01)	14.18 (*p* = 0.11)	9.65 (*p* = 0.77)	34.97 (*p* = 0.35)	65.35 (*p* = 0.18)	95.74 (*p* = 0.14)

## Data Availability

The original contributions presented in this study are included in the article/Appendix A. Further inquiries can be directed to the corresponding author(s).

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
