# Peer review of "Impact of the ABxSG Mobile Application on Antibiotic Prescribing: An Interrupted Time Series Study"

_antibiotics, 2025, doi:10.3390/antibiotics14090933_

Round 1
Reviewer 1 Report
Comments and Suggestions for Authors
I congratulate the authors for their studies evaluating the potential impact of the ABxSG mobile application on antibiotic prescribing. Their work holds promise for the future of antimicrobial stewardship.
You mentioned that an Antimicrobial Stewardship Unit (ASU) was established at Singapore General Hospital (SGH) in 2008. Thirteen years after this unit was established, your 2021 prevalence study (PPS) using the Global-PPS methodology showed that 48% of inpatients on the study day received at least one antibiotic. The ASU, a multidisciplinary committee, has been working effectively for 13 years. However, despite its efforts, approximately 20-25% of antibiotic prescriptions at SGH were deemed inappropriate based on internal observations. Can these results be explained solely by clinicians not referring to the guidelines available in the hospital system?
As of September 2024, you stated that ABxSG has been downloaded 1,902 times. In addition to the distribution of users, it may be appropriate to indicate the percentage of physicians, pharmacists, and nurses who downloaded this program. Since some of the antibiotics used are for prophylaxis, what percentage of surgeons, for example, have downloaded this program?
In addition to the application itself, the authors' efforts in increasing awareness have significantly contributed to the success achieved in antibiotic use following ABxSG implementation. Their work in reducing bias in this regard is commendable.
Beyond your behavioural design-focused approach, have you considered utilising implementation engineering approaches? These approaches could potentially enhance the effectiveness of the ABxSG mobile application.
You mentioned that direct savings in healthcare and labour costs are expected. Could you provide more details or calculations regarding these savings? Additionally, do you have any calculations regarding the cost of producing and maintaining ABxSG's content?
Author Response
Comment 1: You mentioned that an Antimicrobial Stewardship Unit (ASU) was established at Singapore General Hospital (SGH) in 2008. Thirteen years after this unit was established, your 2021 prevalence study (PPS) using the Global-PPS methodology showed that 48% of inpatients on the study day received at least one antibiotic. The ASU, a multidisciplinary committee, has been working effectively for 13 years. However, despite its efforts, approximately 20-25% of antibiotic prescriptions at SGH were deemed inappropriate based on internal observations. Can these results be explained solely by clinicians not referring to the guidelines available in the hospital system?
Response 1: Thank you for your comment. We have edited the manuscript in line 62 to line 66 to emphasize the importance of education and motivation in promoting appropriate antibiotic prescribing.
A survey conducted by Pelullo et al. showed that surgeons who used unnecessary surgical prophylaxis were concerned about patients developing surgical site infections if antibiotics were not given [12]. Hence, apart from ensuring guideline accessibility, availability of effective learning resources and motivating clinicians to refer to them are important factors in advocating appropriate antibiotic prescribing.
Comment 2: As of September 2024, you stated that ABxSG has been downloaded 1,902 times. In addition to the distribution of users, it may be appropriate to indicate the percentage of physicians, pharmacists, and nurses who downloaded this program. Since some of the antibiotics used are for prophylaxis, what percentage of surgeons, for example, have downloaded this program?
Response 2: Thank you for your comment. In line 198 to line 199, we have stated the distribution of users by profession, “As of September 2024, there were 1,902 downloads of ABxSG; 70% of the users were doctors, followed by pharmacists (28%), and nurses (2%).”
We acknowledge that it would have been ideal if we are able to know the department of the users who downloaded the app and which pages they frequently visited to further improve the application. However, we are limited by our application analytics. We have edited the manuscript in line 380 to line 384 to state this as a limitation.
Lastly, due to limitations in the application analytics, we are unable to accurately identify the department that users belong and their frequently visited pages. We acknowledge that understanding the user profile and usage patterns are critical in improving the mobile application. Hence, one important lesson learnt for us would be the importance of building a robust application analytical system following any mobile application development to facilitate ongoing enhancement.
Comment 3: In addition to the application itself, the authors' efforts in increasing awareness have significantly contributed to the success achieved in antibiotic use following ABxSG implementation. Their work in reducing bias in this regard is commendable.
Response 3: Thank you for your encouragement.
Comment 4: Beyond your behavioural design-focused approach, have you considered utilising implementation engineering approaches? These approaches could potentially enhance the effectiveness of the ABxSG mobile application.
Response 4: Thank you for your question.
As the main goal of ABxSG is to change the habits of healthcare providers with respect to antibiotic prescription, we sought to use behavioral design to address clinicians’ concerns and aim to modify their habits. In healthcare mobile applications, success depends heavily on whether clinicians actually find the tool relevant, easy to use and whether they are supportive of the content. Design thinking helps us to identify the right problem and craft user-centric solutions. Hence, behavioral design-focused approach was our priority.
In fact, in the development process, implementation engineering was also adopted in a way that ensures that the app is robust, scalable and functional. For example, (in line 111 to line 115) to allow for content from multiple institutions of the SingHealth hospitals to be hosted distinctly within the same application, we built a containerized infrastructure so that our sister institutions can upload their personalized content in their respective “containers”. This feature supports physicians with cross-institution practices, allowing them to switch “containers” (institutions) easily within the same application.
Comment 5: You mentioned that direct savings in healthcare and labour costs are expected. Could you provide more details or calculations regarding these savings? Additionally, do you have any calculations regarding the cost of producing and maintaining ABxSG's content?
Response 5:
We have indicated the expected cost-savings for antibiotics in line 264 to 267. We have further added in the manuscript regarding the estimated labour cost saved from nursing manpower perspective in line 267 to 269 and cost of developing ABxSG in line 116 to line 118.
“Based on in-house data where an estimated 60% of inpatients on antibiotics are prescribed IV formulation, we project that the monthly IV drug administration cost [approximately SGD 78/day (~USD 60)] avoided with 650 antibiotic-free days is more than SGD $30,000 (~USD23,000). Furthermore, assuming 30 minutes is needed to prepare and administer an IV antibiotic; an estimated 585 hours/month of nursing time can be saved [approximately SGD 17,000 (~USD13,000) a month].
“Unwavering support and buy-in from senior management enabled both funding disbursement [SGD $60,000 (~USD 46,000) for developing ABxSG] and manpower deployment required for the development and maintenance of ABxSG.”
Reviewer 2 Report
Comments and Suggestions for Authors
Thank you for your submission. I enjoyed reading the success of your program.
Can you address:
- the use of proportion of patients on antibiotics and how it does not adjust for the number of patients hospitalized. How do we know if the reduction of antibiotic utilization is not due to a reduction in the number of patients that are undergoing elective surgery or having infections?
- Why appropriateness of antibiotics was not assess at the disease state level (UTI, pneumonia, or pre-operative) or unit/department. It would be good to know where the impact of antibiotic utilization is happening.
- Expand on why there was an immediate reduction in proportion of inpatients on antibiotics following the ABxSF launch despite the relatively low adoption at 3 months (680 users) vs at 18 months (1902 users)
Author Response
Comment 1: Can you address the use of proportion of patients on antibiotics and how it does not adjust for the number of patients hospitalized. How do we know if the reduction of antibiotic utilization is not due to a reduction in the number of patients that are undergoing elective surgery or having infections?
Response 1: Thank you for your question. There is a possibility that there could be minor fluctuations in the characteristics of the patient population admitted from month to month. However, our team believes that using proportion of patients on antibiotics is an appropriate approach of normalizing the data to ensure fair comparison across data points. To our knowledge, we are not aware that there is a drastic change in the characteristics of our admitting criteria/patient population. For example, the percentage of admission for elective surgeries were stable throughout the 3-year study period, ranging from 16 to 22%, with minor fluctuations in some months.
With regard to number of patients having infections, trending it may not yield meaningful results. For example, a reduction in number of patients diagnosed with a particular infection may suggest that our antibiotic stewardship efforts are working well by reinforcing the importance of appropriate diagnosis of infection over time and may not be because the patient population characteristics have changed over time.
Comment 2: Why appropriateness of antibiotics was not assess at the disease state level (UTI, pneumonia, or pre-operative) or unit/department. It would be good to know where the impact of antibiotic utilization is happening.
Response 2: Thank you for your question. This is a very relevant question.
We did consider breaking down our analysis further into antibiotic indication. However, by doing so, it will result in very small denominators, which are susceptible to fluctuations and hence risk, not being able to identify any meaningful trends. We have included this under our study limitations in the following line 375 to line 379.
Thirdly, the appropriateness of antibiotic was not assessed based on indication as data pertaining to smaller groups are susceptible to fluctuations due to small denominators and hence, unlikely to identify any meaningful trends. Nonetheless, we noticed that there were no major differences in the type of antibiotic inappropriateness (based on choice, duration, route, dosing, need for antibiotic) over time and we believe that improvement in antibiotic prescribing likely applies to all categories of indication in general.
Comment 3: Expand on why there was an immediate reduction in proportion of inpatients on antibiotics following the ABxSG launch despite the relatively low adoption at 3 months (680 users) vs at 18 months (1902 users)
Response 3: While 680 users at 3 months is considered small compared to 1902 users at 18 months, we estimate that about 450 (out of 680 users) are doctors, representing about 30% of doctors in the institution (a significant proportion). With the launch of ABxSG, doctors now have ready access to antibiotic guidelines with clinical pearls on antibiotic use. On-call doctors can use the information to determine if they should start a febrile patient on antibiotic or consider other differential diagnoses. For surgeons, the guidelines also provide recommended duration for surgical prophylaxis, reducing excessive duration of prophylaxis in clean surgeries.
Reviewer 3 Report
Comments and Suggestions for Authors
Lee et al developed a mobile application and studied its impact on antibiotic prescription. I congratulate the authors on this important and interesting work.
I have a few minor points:
- The title is too complex (in my modest opinion). I suggest eliminating the name of the application from the title, ABXSG. Moreover, why not just “Impact of a Mobile Application on Antibiotic Prescribing”? I don’t feel one needs to know the method in the title by mentioning the interrupted time series. This is just one suggestion, but, of course, I leave that issue to the authors and editor.
- I understand researchers studied the prescription of these three antibiotics just as an example. But I would suggest some more information about these antibiotics. Why did the authors focus on these antibiotics?
- Moreover, what is the resistance level (percentage) to these three antibiotics of pathogens in Singapore? If this information is available, it would be interesting to the reader.
- Concerning Table 1, I understand that showing “Change in slope” is interesting, but I would suggest some discussion on the relationship between the change in slope and what happens at 1-month, …, 18-month. For example, for Piperacillin, the slope is positive, but it is negative for the other two antibiotics. What does that imply, or what is the cause of that difference? Anyway, p-values are always above 0.05, so perhaps one cannot extract interesting information from these differences. Could the authors comment on these issues?
- There are some problems with the format. For example, starting on line 217, the text is in bold.
- Line 235: Instead of using the phrase “was stable”, consider changing to something like “did not change”.
- Starting in line 244, there is data concerning “pharmacist-led interventions”. Does that mean that some prescriptions were done by the pharmacist, and not by a medical doctor? A few words on this would be useful for some interested readers not working in hospitals (and not from Singapore). These things may differ between countries, I think.
- Consider lines 209-210 “As of September 2024, there were 1,902 downloads of ABxSG; 70% of the users were doctors, followed by pharmacists (28%), and nurses (2%).” Why such differences? Does that mean that the same proportion of professionals adhered to the app ABXSG? Can nurses also prescribe antibiotics? And what about pharmacists? (For example, in my country, only medical doctors can prescribe antibiotics).
- SGH stands for what? Singapore hospitals? Just one hospital?
- In the discussion, please explain the meaning of (e.g., line 270) “antibiotics are on IV formulation”.
- Line 276: Consider changing from “one of the most” to “two of the most”
- Line 296: Consider changing from “select” to “selected”
Author Response
Comment 1: The title is too complex (in my modest opinion). I suggest eliminating the name of the application from the title, ABXSG. Moreover, why not just “Impact of a Mobile Application on Antibiotic Prescribing”? I don’t feel one needs to know the method in the title by mentioning the interrupted time series. This is just one suggestion, but, of course, I leave that issue to the authors and editor.
Response 1: Thank you for your suggestion. We would prefer to keep to our current title.
Comment 2: I understand researchers studied the prescription of these three antibiotics just as an example. But I would suggest some more information about these antibiotics. Why did the authors focus on these antibiotics?
Response 2: Appropriateness of antibiotic data is only available for antibiotics that are audited by prospective audit feedback by antimicrobial stewardship unit pharmacists. Out of those that we audit, piperacillin-tazobactam, ceftriaxone and meropenem make up more than 90% of the audits. Hence, piperacillin-tazobactam, ceftriaxone and meropenem appropriateness data were analyzed.
We have improved the clarity in our manuscript in line 180 to line 185.
Notably, meropenem, piperacillin-tazobactam and ceftriaxone contribute to 90% of audits conducted by ASU pharmacists and are important workhorses in the antibiotic armamentarium, whose susceptibility must be preserved. Based on aggregated data from acute care hospitals in Singapore in 2021, resistance to ceftriaxone ranges from 21.0% for Escherichia coli to 23.3% for Klebsiella pneumoniae. For meropenem, resistance ranges from 0.3% for Escherichia coli to 31.5% for Acinetobacter baumanii [15].
Comment 3: Moreover, what is the resistance level (percentage) to these three antibiotics of pathogens in Singapore? If this information is available, it would be interesting to the reader.
Response 3: We have improved the clarity in our manuscript in line 180 to line 185.
Notably, meropenem, piperacillin-tazobactam and ceftriaxone contribute to 90% of audits conducted by ASU pharmacists and are important workhorses in the antibiotic armamentarium, whose susceptibility must be preserved. Based on aggregated data from acute care hospitals in Singapore in 2021, resistance to ceftriaxone ranges from 21.0% for Escherichia coli to 23.3% for Klebsiella pneumoniae. For meropenem, resistance ranges from 0.3% for Escherichia coli to 31.5% for Acinetobacter baumanii [15].
Comment 4: Concerning Table 1, I understand that showing “Change in slope” is interesting, but I would suggest some discussion on the relationship between the change in slope and what happens at 1-month, …, 18-month. For example, for Piperacillin, the slope is positive, but it is negative for the other two antibiotics. What does that imply, or what is the cause of that difference? Anyway, p-values are always above 0.05, so perhaps one cannot extract interesting information from these differences. Could the authors comment on these issues?
Response 4: The definition of the change in slope is explained in line 191 to line 192 “Pre-slope was the trend observed prior to ABxSG whereas slope change refers to the change in direction and gradient of the trend line after ABxSG launch.” and also under the footnote of Table 1. We did not comment about the slopes as the changes were not statistically significant and hence unable to conclude if change in slope direction was indeed meaningful.
Comment 5: There are some problems with the format. For example, starting on line 217, the text is in bold.
Response 5: We have rectified the issues.
Comment 6: Line 235: Instead of using the phrase “was stable”, consider changing to something like “did not change”.
Response 6: We have rectified the issue.
Comment 7: Starting in line 244, there is data concerning “pharmacist-led interventions”. Does that mean that some prescriptions were done by the pharmacist, and not by a medical doctor? A few words on this would be useful for some interested readers not working in hospitals (and not from Singapore). These things may differ between countries, I think.
Response 7: We have added the following paragraph to line 240 to line 243 for better clarity:
At SGH, all inpatient antibiotic orders are verified by pharmacists before they are administered to the patients and continued use of antibiotics are regularly reviewed by either ward pharmacists or ASP pharmacists. If the antibiotics are deemed inappropriately used at any time point, an intervention will be made by a pharmacist to modify antibiotic order.
Comment 8: Consider lines 209-210 “As of September 2024, there were 1,902 downloads of ABxSG; 70% of the users were doctors, followed by pharmacists (28%), and nurses (2%).” Why such differences? Does that mean that the same proportion of professionals adhered to the app ABXSG? Can nurses also prescribe antibiotics? And what about pharmacists? (For example, in my country, only medical doctors can prescribe antibiotics).
Response 8: In Singapore, only doctors can prescribe antibiotics which explains why the bulk of the users are doctors as the goal of the app is to ensure appropriate antibiotic prescribing. However, pharmacists and nurses are also involved in care of patients on antibiotics and there are information pertaining to each healthcare professions such as dosing for pharmacists and administration pointers for nurses. Also, the number of pharmacists to doctors in SGH is about 1:10.
We have also added the following sentence to emphasize that antibiotic prescribing is done by medical doctors at our institution in line 199 to line 202.
At SGH, antibiotics can only be prescribed by doctors, which makes them the primary target audience for ABxSG. However, as pharmacists and nurses are also involved in the care of patients on antibiotics, the app also includes information that aid those healthcare professionals.
Comment 9: SGH stands for what? Singapore hospitals? Just one hospital?
Response 9: SGH refers to Singapore General Hospital.
Comment 10: In the discussion, please explain the meaning of (e.g., line 270) “antibiotics are on IV formulation”.
Response 10: The statement refers to that 60% of all antibiotics prescribed in SGH are given intravenously. We have improved the clarity of this sentence by rephrasing as follows in line 264 and line 267:
“Based on in-house data where an estimated 60% of inpatients on antibiotics are prescribed IV formulation, we project that the monthly IV drug administration cost [approximately SGD 78/day (~USD 60)] avoided with 650 antibiotic-free days is more than SGD $30,000 (~USD23,000).
Comment 11: Line 276: Consider changing from “one of the most” to “two of the most”
Response 11: We have rectified the issue.
Comment 12: Consider changing from “select” to “selected”
Response 12: We have rectified the issue.